# Mindfulness-Enhanced Computerized Cognitive Training for Depression: An Integrative Review and Proposed Model Targeting the Cognitive Control and Default-Mode Networks

**DOI:** 10.3390/brainsci12050663

**Published:** 2022-05-19

**Authors:** Mikell Bursky, Dakota A. Egglefield, Sophie G. Schiff, Pranitha Premnath, Joel R. Sneed

**Affiliations:** 1Psychology Department, The Graduate Center, City University of New York, New York, NY 10016, USA; degglefiel@gradcenter.cuny.edu (D.A.E.); sschiff@gradcenter.cuny.edu (S.G.S.); ppremnath@gradcenter.cuny.edu (P.P.); joel.sneed@qc.cuny.edu (J.R.S.); 2Psychology Department, Queens College, City University of New York, Queens, New York, NY 11367, USA

**Keywords:** computerized cognitive training, mindfulness, depression, cognitive control network, default-mode network

## Abstract

Depression is often associated with co-occurring neurocognitive deficits in executive function (EF), processing speed (PS) and emotion regulation (ER), which impact treatment response. Cognitive training targeting these capacities results in improved cognitive function and mood, demonstrating the relationship between cognition and affect, and shedding light on novel targets for cognitive-focused interventions. Computerized cognitive training (CCT) is one such new intervention, with evidence suggesting it may be effective as an adjunct treatment for depression. Parallel research suggests that mindfulness training improves depression via enhanced ER and augmentation of self-referential processes. CCT and mindfulness training both act on anti-correlated neural networks involved in EF and ER that are often dysregulated in depression—the cognitive control network (CCN) and default-mode network (DMN). After practicing CCT or mindfulness, downregulation of DMN activity and upregulation of CCN activity have been observed, associated with improvements in depression and cognition. As CCT is posited to improve depression via enhanced cognitive function and mindfulness via enhanced ER ability, the combination of both forms of training into mindfulness-enhanced CCT (MCCT) may act to improve depression more rapidly. MCCT is a biologically plausible adjunct intervention and theoretical model with the potential to further elucidate and target the causal mechanisms implicated in depressive symptomatology. As the combination of CCT and mindfulness has not yet been fully explored, this is an intriguing new frontier. The aims of this integrative review article are four-fold: (1) to briefly review the current evidence supporting the efficacy of CCT and mindfulness in improving depression; (2) to discuss the interrelated neural networks involved in depression, CCT and mindfulness; (3) to present a theoretical model demonstrating how MCCT may act to target these neural mechanisms; (4) to propose and discuss future directions for MCCT research for depression.

## 1. Introduction

Major depressive disorder (MDD) is a psychiatric condition that affects an estimated 163 million people (2% of the world’s population) and is associated with reduced quality of life, educational attainment, productivity, disability and suicide [1,2,3,4,5]. MDD is a heterogeneous syndrome consisting of many depressive subtypes such as melancholia subtype, atypical subtype, post-stroke depression and vascular depression, as well as less severe variants such as persistent depressive disorder [6,7,8]. Although based on correlational data, the past decade has witnessed a 33% rise in the overall incidence and prevalence of MDD, which may have been driven by factors such as greater awareness of mental health, increased treatment options, de-stigmatization of psychiatric conditions and increased use of electronics and social media [9,10]. More recently, the COVID-19 pandemic has added additional societal-level stressors associated with increased depression rates [11]. The prevalence of depressive symptoms in the U.S. increased by more than 3-fold, from 8.5% before the emergence of COVID-19 to 27.8% during the ongoing public health crisis [12]. Moreover, between 2009 and 2018, anti-depressant use among U.S. adults increased from 10.6% to 13.8% [13]. The alarming increase in depression rates has contributed to an increased burden on an already overwhelmed healthcare system, necessitating further research into new and effective interventions [14].

MDD is characterized by depressed mood and difficulties with emotion regulation (ER), which contribute to commonly observed affective symptoms, such as persistent feelings of sadness, amotivation, irritability, anhedonia, rumination, emptiness, hopelessness and suicidal ideation [2,6,15]. Behavioral symptoms commonly include sleep disturbances, reduced appetite, self-harm and suicide [2,3]. Further, neuropsychological deficits in executive functions (EF) and processing speed (PS) often co-occur with affective dysregulation and contribute to the depressive syndrome [15,16,17,18,19,20]. Moreover, depressive symptoms can also impact an individual’s daily functioning, such as their ability to work, complete their education and socialize [1,2,3,5]. Lastly, severe and chronic symptoms can lead to physical disability, illness and death [4].

Psychiatric medications such as selective serotonin reuptake inhibitors (SSRIs) and serotonin–norepinephrine reuptake inhibitors (SNRIs) have traditionally been the first line of treatment for MDD in conjunction with psychotherapy [21]; however, side effects from such medications are common and make continuation and adherence to treatment difficult [22,23]. Psychotherapy (e.g., cognitive behavioral therapy, behavioral activation, short-term psychodynamic psychotherapy, interpersonal psychotherapy) constitutes a second line of treatment but it can be time-consuming, require extensive effort or be unavailable to patients [24,25,26,27,28]. Although the relationships between the additive effects of psychotherapy and medication are complicated, psychotherapy has been found to be an effective treatment, either alone or in combination with medication for a variety of depressive subtypes [29]. There is also evidence that the effects of psychotherapy alone may be overestimated but nevertheless positive. For example, a meta-analysis by Cuijpers et al., (2010) found the effects to be overestimated after controlling for study quality (i.e., *d* = 0.22 for high quality compared to *d* = 0.74 for lower quality) [30]. When combined, SSRIs and psychotherapy tend to produce the best results [31,32]. Unfortunately, only 60% of individuals achieve remission after their first use of medication [33] and at least one-third of individuals treated with either psychotherapy or medication do not respond adequately, despite multiple attempts [34]. Other treatments such as electroconvulsive therapy or deep brain stimulation may be low in feasibility or invasive [35]. Moreover, none of these treatments target the cognitive deficits, which often co-occur in depression and impact treatment response [36,37,38,39]. As such, alternative treatment options that target these deficits should be investigated.

Computerized cognitive training (CCT) is one such alternative treatment that has received attention [40,41,42,43]. CCT consists of software-based exercises that target, train and strengthen specific cognitive processes through repeated practice [44]. However, it should be noted that controversy surrounding the use of CCT and its effectiveness as an intervention for various patient populations has been met with debate [44]. While proponents of CCT point to systematic reviews showing improved cognitive abilities and some evidence of these benefits transferring over into daily functioning [45], others contend that manufacturer claims of improved mental fitness through CCT are overstated and potentially misleading to vulnerable patient populations and consumers [46]. Further, critics highlight the paucity of methodologically rigorous evidence to support the claim that the cognitive improvements generalize more broadly or whether the improved performance observed does not actually reflect the similarity between the training paradigms and the neuropsychological measures used in research [44,47].

Nonetheless, there is evidence to suggest that CCT may be effective in the treatment of depression and other disorders through targeting and training deficient underlying cognitive processes [43,47,48,49]. It is not yet clear whether CCT can be used as a standalone treatment, but there is evidence that CCT can be used as an adjunctive treatment, which may act to boost psychotherapy or medication response [50,51,52]. The use of CCT combined with behavioral activation methods for depression, however, has not shown any beneficial effects above and beyond behavioral activation alone [53]. In comparison with psychotherapy or medication, CCT has important advantages, such as being relatively inexpensive, convenient, quick, non-invasive, personalized, private and accessible, while lacking side effects and reducing stigma [47,54].

Although CCT targets aspects of cognition, one relative disadvantage compared to other forms of training, such as mindfulness, is that CCT does not directly target affective dysregulation and negative self-referential processing, which are hallmarks of depression. Further, CCT does not provide ER methods or train an individual to regulate difficult emotions. Rather, CCT focuses on improving underlying cognitive factors necessary for the successful execution of ER, such as the ability to inhibit maladaptive thought processes and shift attention away from depressive rumination. As such, the combination of CCT, which targets cognition directly, with another type of intervention that targets ER and self-referential processing may work to boost treatment response.

One parallel line of research in novel treatments for depression, with a specific focus on improving ER and self-referential processes, is that of mindfulness [55,56]. Mindfulness is a form of mental training that helps an individual learn to regulate their attention, emotions and mental processes while maintaining a relaxed, non-judgmental state of mind [57]. Various durations of mindfulness practice have been found to enhance ER capability, improve cognitive abilities and augment self-referential processing [58,59,60]. A growing body of evidence suggests that mindfulness interventions are efficacious in improving mood and treating depression [61,62]. Mindfulness training has also been successfully combined with cognitive therapy to form mindfulness-based cognitive therapy (MBCT) to treat severe depression [63]. MBCT has been found to cut relapses in recurrent major depression more effectively than cognitive therapy or anti-depressant medication [64,65]. More recently, MBCT has been developed in a computerized format and shown promising benefits for pain management [66]. MBCT or its online version teach patients to become aware of thoughts through the practice of mindfulness and how to restructure maladaptive cognitive processes associated with depressive symptoms [63]. It does not, however, specifically target, train and strengthen underlying neurocognitive processes as occurs with CCT [44,67].

While CCT is theorized to target cognitive deficits and brain networks that underlie them, such as the cognitive control network (CCN), mindfulness may target both cognition and ER, as well as neural networks thought to be implicated in depression, namely the default-mode network (DMN) [68,69,70,71]. Moreover, activity in the CCN and DMN is anti-correlated and differentially augmented by mindfulness and CCT [43,60,68,72,73,74]. As such, the union of these two forms of training may act to target these networks and more rapidly improve depression. Based on these findings, we propose that the next step in the progression of CCT is to explore the potential benefits of combining mindfulness training with a regimen of CCT in order to boost treatment response in depression: We propose mindfulness-enhanced CCT (MCCT).

The aims of this integrative review article are four-fold: (1) to briefly review the current evidence supporting the efficacy of CCT and mindfulness interventions in improving affective and cognitive symptoms associated with depression; (2) to discuss the interrelated neural networks implicated in depression; (3) to present a biologically plausible model demonstrating how MCCT may target these neural mechanisms; (4) to discuss and propose the feasibility, transfer of benefits and future directions for MCCT research.

## 2. Depression and Computerized Cognitive Training

Individuals with depression often experience debilitating co-occurring cognitive and affective deficits, which include reduced EF, PS and ER [15,16,17,19,20,37,71,75,76,77]. EF consists of cognitive control mechanisms tasked with the activation and coordination of multiple cognitive subprocesses, such as attention and working memory, as well as the updating of goal-relevant information and the inhibition of goal-irrelevant information [39,78]. When interference and distraction from goal-irrelevant information are present, EF allows an individual to inhibit this interference and maintain goal-relevant information in working memory [79]. PS is considered a lower-order function that allows EF to work effectively [80,81], and as such may be a crucial target for cognitive-focused interventions [49,67,82]. Deficits in EF and PS observed among depressed individuals diminish their ability to regulate emotion and respond to treatment [37,83,84,85]. Further, these deficits contribute to various cognitive symptoms, such as difficulty processing information, thinking, concentrating, decision-making and remembering [15,16,17,18,19,20]. Cognitive deficits among depressed older adults are common, with more than 40% also experiencing executive dysfunction [86,87]. Moreover, older adults with executive dysfunction are two times less likely to respond to anti-depressant treatment [88]. In fact, the degree to which cognitive impairment is observed in EF and PS is the strongest predictor of functional recovery and treatment response in major depression [36,38]. The co-occurring executive dysfunction and decreased PS often found in depressed individuals highlights the importance of addressing these deficits [37,39].

One intervention that may help address these deficits is CCT. CCT consists of cognitive exercises or games that target and strengthen specific neural networks to improve cognitive functioning through the principles of neural plasticity [44,50]. To begin CCT, an individual can utilize any technology in which a particular CCT program is available, such as on a smartphone, iPad or computer. After gaining access to a particular CCT program, an individual will begin playing different brain training games for any chosen amount of time based on their personal interests. To make responses during a game, an individual engages with the user interface specific to the CCT platform. Responses generally consist of pressing buttons on a screen, however, more advanced video-game-style CCT interventions are also becoming available [89]. Different game modules train various domains of cognitive function such as working memory, attention, processing speed and visuo-spatial mapping through repeated practice. Some games focus on practical skills such as improving reaction times and sustained attention related to driving, while others focus on memory skills related to names or to-do lists. Algorithms track and monitor progress, tailoring the difficulty level based on performance, which is consistent with the goals of personalized medicine [90]. In this way, CCT games become more challenging as one’s abilities and performance improve.

Due to the brain’s neuroplastic quality, training cognitive abilities through repeated CCT exercises can lead to improvements in both the structural integrity and functional connectivity in the CCN, which in turn may lead to enhanced PS, improved EF and improvements in depressive symptoms [45,51,67,82,89,91]. One proposed model of action places PS as a hypothesized mediator between engagement in CCT and improved EF [67]. As improved EF is associated with increased regulation of negative affect, this may lead to the alleviation of depressive symptoms [67,82].

CCT has shown to be effective in improving depressive symptoms and cognitive functions in younger, middle-aged and older adults, both in combination with psychotherapy and medication [45,47,89,92]. In the event of treatment non-response to medication [33], CCT may be implemented as a complementary or adjunct treatment option that enhances psychotherapy and medication (e.g., boosts SSRI response) [67,93]. Studies of CCT focused on enhancing cognitive control and PS among depressed older adults have found improvements in multiple domains of EF and depressive symptoms [45,49,51,91,92]. Among younger adults with mild depression, CCT focused on EF or PS training likewise found improvements in mood and cognitive function [43]. Lastly, Gunning et al., (2021) tested a video-game-style CCT intervention among middle-aged and older adults diagnosed with MDD [89]. The study found improvements in cognitive control functions such as sustained attention and working memory, as well as improvements in apathy and depressive mood symptoms [89].

## 3. Depression and Mindfulness Training

Over the past decade, researchers and clinicians have taken a keen interest in the application of various forms of mindfulness-based interventions (MBIs) to mitigate a range of psychopathological symptoms [55,64,94]. Mindfulness can be understood as a non-judgmental, accepting and open awareness of all thoughts, feelings, sensations and perceptions experienced in the present moment, which is sustained throughout daily activities via the continual practice of mindfulness–meditation [58,59,95]. In this sense, mindfulness is multifaceted; it is both a meditative technique, a state of awareness and a psychological trait that can be enhanced through training [59]. More recently, meditation and mindfulness-based techniques have been categorized into three main types, understood as being on a spectrum ranging from conceptual, constructive and self-referential, to non-conceptual, deconstructive and self-less awareness, namely focused attention (FA), open monitoring (OM), and non-dual awareness (NDA) meditation, respectively [96,97].

Mindfulness techniques and protocols often vary between studies, making it difficult to achieve a standardized protocol (e.g., different mindfulness methods, instructors, instructions, durations and intervention fidelity) [59]. To help clarify the types of mindfulness protocols implemented in the cited studies and to provide examples as to what these methods of mindfulness training entail, a brief overview is now provided.

Mindfulness training drawn from Buddhist meditative traditions is often divided into two main stages—the first focuses on the development of concentration or mental tranquility, while the second focuses on the development of insight into the nature of one’s thought processes and reality itself [98]. These two stages map onto FA and OM, respectively, with NDA being the most advanced end of the insight spectrum [96].

Formal mindfulness training methods are generally practiced in a state of physical stillness while seated in an upright position; however, postures can vary and be adjusted based on one’s abilities and needs (e.g., sitting in a chair, legs crossed, partially crossed legs, etc.) [99]. The eyes are sometimes closed for different techniques; in particular, for beginning concentration techniques that utilize FA, the eyes may be partially closed to help calm the mind while maintaining wakefulness. Other practices such as OM and NDA often utilize open eyes and are considered more advanced methods [99]. In more later stages of mindfulness training, the practice is ultimately meant to become integrated with one’s daily activities, such that mindful awareness and its associated qualities are brought to bear in all aspects of one’s life, not only during formal training periods [100]. From the perspective of modern cognitive science, the practice of mindfulness is thought to enhance meta-awareness or meta-cognition, the ability to be consciously aware of one’s current thought processes [101].

FA methods refer to practices in which an individual directs their attention in a more narrowed and concentrated way to a specific object [96]. For example, attention may first be directed towards one’s breath; anytime focus is lost and the mind wanders, the practitioner learns to recognize that their mind has wandered and refocuses their attention back onto their breath [102]. With continued practice, concentration develops and attention can be focused on a single chosen object for a protracted amount of time without losing clarity [98]. FA methods can also include focusing attention on outer objects such as a candle, a flower, a statue, physical sensations or mantras (sacred syllables), or on self-visualized objects [99].

OM methods consist of practices in which attention is no longer focused on a single object, rather one’s attentional scope is expanded into a natural awareness that allows all thoughts, feelings, sensations and perceptions to arise in one’s experience without judgment or reactivity [103]. A practitioner simply monitors and is aware of whatever happens to arise in their mind or in the world around them, without judgment, attachment or aversion [104].

NDA methods are the least tangible in terms of understanding them conceptually. Practices that emphasize this type of training are found in the Tibetan Buddhist traditions of Dzogchen and Mahamudra, the Zen tradition and the Hindu practice of Advaita Vedanta [97]. Josipovic (2010) describes NDA as a unified state of pure consciousness in which the ordinary experience of subject and object is no longer experienced in a fragmented or dualistic way [105]. Through these NDA practices, one’s ordinary sense of self may transform into a state of non-reactive pure awareness of the present moment [106].

Meditation and mindfulness-based practices have been shown to improve cognitive function as well as depressive symptoms. Improvements in visuo-spatial processing, EF and mood have been found after as little as 4 days of practice [57]. Tang and Posner (2014) propose that executive neural networks can be trained to function more efficiently using both cognitive tasks and meditation training [48]. Long-term meditation training, for example, may help mitigate neurocognitive deficits often observed in older adults, as evidenced by improvements in neuropsychological functioning post-training [107,108]. This may be particularly relevant for older adults with treatment-resistant late-life depression (LLD), in which deficits in EF and PS are common [109]. Further, lonely older adults who are at higher risk of developing depression and mortality [110,111] have shown improvements in well-being, reductions in loneliness and interestingly reductions in pro-inflammatory gene expression associated with loneliness through the practice of mindfulness [112]. Moreover, after practicing mindfulness, depressive symptoms associated with self-rumination have shown improvement [113].

More recently, research on smartphone-based mindfulness meditation applications have demonstrated beneficial effects on depressive symptoms and well-being [62,114]. MBIs have traditionally been administered in person and in group settings; however, with the rapidly changing social landscape, smartphone-based MBIs are worth exploring. Although smartphone-based MBIs have shown smaller effect sizes on psychological outcomes compared to when delivered in their original in-person formats, improvements in psychological functioning have still been observed, show promise and provide a wider range of accessibility [114,115,116,117].

Although the results related to the effects of MBIs on psychological outcomes are encouraging, further research is warranted. For example, Wang et al., (2018) conducted a meta-analysis examining the effects of MBIs on MDD in randomized controlled trials (RCT) [56]. They found that MBIs conducted for a short duration of time do not yield long-term benefits; rather, long-term gains may be dependent on the continual practice of mindfulness–meditation over longer intervals of time [56]. As such, future interventions should consider using larger samples with trials of longer duration.

## 4. Neural Networks Involved in Depression

Depression has been found to be marked by decreased activation and lower resting-state functional connectivity in the CCN, as well as increased activation and higher resting-state functional connectivity in the DMN [68,73,75,118,119,120,121,122,123]. The CCN is an externally oriented brain network consisting of cortico-limbic circuitry that is often found impaired in individuals with depression [124,125,126]. The DMN is an internally-oriented network of brain regions involved in internal modes of cognition, and is often dysregulated in depression [75,127]. Among depressed individuals, the DMN is generally hyperactivated and associated with self-rumination and mind wandering [71,128], while deficits in information processing speed and cognitive control have been attributed to dysfunction in the CCN [125,129,130]. Dysfunction within the CCN and DMN has been implicated in many of the common behavioral symptoms observed in individuals with depressive disorder, such as apathy, anxious depression and suicidality [123]. Among MDD patients as compared to healthy controls, anti-correlated resting-state functional connectivity between the CCN and DMN, as well as weakened functional connectivity within the CCN itself, has been observed and predictive of depressive symptom severity [131].

It is important to note that nearly all aspects of cognitive function are brought about through the integration of distributed networks of neural activity [132]. For example, more recent accounts of cognitive control show distributed whole-brain neural network involvement, while earlier modular-based explanations focused solely on attention and conflict monitoring in the dorsolateral prefrontal cortex (DLPFC) and anterior cingulate cortex (ACC), respectively [133,134,135]. The dynamic coupling between functional brain networks suggests that many anatomical networks are interrelated and are often connected by structural brain hubs that mediate the neural integration of information [132,136].

In relation to depression and EF, interconnected networks shed further light on the close relationship observed between cognition and emotion [137,138]. For example, the CCN and DMN are interrelated and anti-correlated; when the CCN is activated, DMN activity is found to be downregulated and associated with depression improvement [71,72,73]. As dysregulation in the CCN and DMN has been associated with depressive symptoms and cognitive impairment, these networks appear to be crucial targets for effective interventions.

### 4.1. Cognitive Control Network

The CCN is normally activated when engaged in goal-directed activity and is involved in the control of executive functions, information processing [139], emotional processing [140] and error detection and resolution [124,129,141,142]. The CCN is composed of two subnetworks, the frontoparietal network and cingulo-opercular network, as well as subcortical limbic structures, which include the thalamus and basal ganglia [125,130]. Among depressed individuals, CCN dysfunction is common and associated with deficits in PS and cognitive control [125,129,130]. In depression, resting-state functional connectivity within the CCN is often reduced relative to healthy controls [143], predictive of depression severity [144] and anti-depressant treatment non-response [68,74]. Further, successful anti-depressant treatment response has been associated with improved functional activation within nodes of the CCN [140]. Moreover, a dysregulated functional interaction between the CCN and limbic or affective circuitry among individuals with MDD has been observed, highlighting the relationship between cognition and affect [138].

To help illustrate the functional relationship between cognition and emotion, Ochsner et al., (2012) provided a model demonstrating the convergence between the CCN and ER mechanisms [126]. This integrated cognitive control–ER system is composed of the ACC, which is tasked with conflict monitoring; the DLPFC and posterior PFC, which are implicated in selective attention and working memory; the ventral lateral PFC, which aids in selection and inhibition; as well as regions involved in subsequent value judgments related to the current stimulus being appraised (i.e., ventral medial PFC and medial orbital frontal cortex). Limbic areas are also activated during the cognitive control of emotion, and are implicated in the encoding of the reward values of the stimulus (i.e., ventral striatum), the arousal or threat value of the stimulus (i.e., amygdala) and finally the subsequent representation of bodily feeling states associated with the experienced emotion in the anterior insular cortex [125,126]. In this way, cognitive and affective processes interact and activate multiple subprocesses involved in the generation of emotions.

Further, the ACC and anterior insular cortex have projection cells that connect to areas involved in the control of both positive and negative emotions [145]. As the ACC is vital for self-regulation and the monitoring of action, this connection serves to function as an interface between affect and cognition, signaling areas of the PFC to increase cognitive control as needed [133,145]. Interestingly, a recent study by Salehinejad et al., (2017) observed that stimulation of the DLPFC using transcranial direct current stimulation improved cognitive function and ER in patients with MDD [146].

### 4.2. Default-Mode Network

The DMN refers to interrelated brain regions involved in internal modes of cognition related to automatic self-referential processing and internal mentation [127]. These processes include autobiographical memory retrieval and self-relevant mental simulations of possible future events [72,147]. The brain regions that comprise the DMN can be subdivided into multiple interacting subsystems, each involved in a different aspect of self-referential processing: the medial temporal lobe subsystem facilitates mental simulation of different events through its provision of self-relevant memories; the medial PFC subsystem facilitates this process through the flexible use of these memories and their associations to create a mental simulation that is self-relevant; finally, these two subnetworks are integrated together at a node in the posterior cingulate cortex (PCC), which connects to the limbic system [72].

While the brain is not engaged in any activity and attention is withdrawn from external events, these regions automatically activate rather than quiet down; conversely, they subside when attention is directed to external stimuli or a cognitive task [73,118,127]. Studies have found that when individuals are not engaged in a task, self-report indicates that their minds wander onto other topics that focus on their own sense of self in relation to the world (i.e., self-referential processing) [148]. Although research findings related to the DMN have been subject to scrutiny, the overall consensus is that the DMN is implicated in these automatic, internally directed self-referential processes [118].

### 4.3. Effects of CCT on Neural Networks

Various forms of CCT target and train the functional neural networks involved in EF, PS and ER [67]. Neurologically, improvements in EF and PS via CCT are thought to be accomplished by directly targeting the CCN [124,125,126]. Through repeated training in CCT, increased experience may drive learning-induced neural plasticity within the CCN, leading to potential improvements in PS, EF and ER capacity [44,51,67,92,149]. The limbic system connections found in the CCN highlight how the dopaminergic reward system may be activated through CCT performance, driving and increasing learning-induced functional plasticity [125,149]. Most recently, Gunning and colleagues (2021) tested the effects of a video-game-style CCT intervention among 34 middle-aged and older adults with a diagnosis of MDD, specifically targeting dysfunction within the CCN [89]. After completing the 4 week intervention, which consisted of 20–25 min of CCT 5 days/week, 74% of the participants showed increased CCN activation during a task and 72% showed increased resting-state functional connectivity associated with improvements in cognitive function and mood [89]. This pattern of increased functional activation and resting-state connectivity within the CCN may be indicative of successful cognitive and affective remediation training [150].

Relatedly, one neurological marker indicative of a cognitive control deficit is decreased activation in midline frontal theta power and frontal–posterior theta coherence (a measure of functional connectivity) when individuals are engaged in tasks of cognitive control [151,152,153]. CCT interventions have yielded increased midline frontal theta activity in older adults, suggesting that the prefrontal cognitive control system in older adults can be remediated to perform more efficiently through cognitive training [154].

Thus far, research into the neural mechanisms of CCT on depression have primarily focused on the CCN; they have not yet fully explored the involvement of the DMN in alleviating symptoms of depression or how these two networks interact in the process. Studies by Meltzer et al., (2007) [155] and Scheeringa et al., (2008) [156] observed decreased DMN activity associated with increased midline frontal theta power among participants engaged in goal-directed activity requiring cognitive control, which suggests that the DMN itself may function within the theta frequency band and be augmented by cognitive training.

### 4.4. Effects of Mindfulness on Neural Networks

The practice of mindfulness meditation is a form of psychological training that augments the CCN and DMN as an individual learns to regulate the focus and quality of their attention [57,60,70,102,119,157]. Although both networks are activated during mindfulness meditation, in this section we first examine studies that assessed changes in the DMN alone, and then studies that assessed changes in both the DMN and CCN together.

In relation to the effects of mindfulness on the DMN, there are numerous noteworthy studies that highlight how mindfulness practice can alter DMN activity. Garrison et al., (2013) used real-time fMRI neurofeedback among a group of meditators and observed that the PCC (a key node in the DMN) showed deactivation when meditators reported an experience of effortless and undistracted awareness [121]. Conversely, subjective reports of effortful and distracted awareness were correlated with increased activation in the PCC [121]. Further, a series of studies by Berkovich-Ohana and colleagues [69,158] revealed 3 main effects of long-term meditation on the DMN: (1) functional connectivity between the DMN and visual networks is higher among non-meditators compared to meditators; (2) functional connectivity within and between the DMN and visual networks is downregulated during meditation compared to resting-state; (3) higher levels of meditation or mindfulness expertise are negatively correlated with DMN functional connectivity. Additionally, research on the effects of meditation on midline frontal theta power and coherence have likewise observed enhanced theta band activity in frontal attention-related areas along with anti-correlated deactivation in parietal–occipital areas associated with self-referential processing or DMN [159,160].

To further explain how meditation and mindfulness may act to alter DMN activity, Laukkonen and Slagter (2021) recently proposed an integrated predictive processing theory of meditation, describing how internal models of the predicted self and world (which are generated in part by the DMN) are deconstructed through the practice of meditation [97]. As individuals focus more completely on the present moment through engagement in meditation, the temporal nature of these internal models is reduced, such that conceptual thoughts related to the past, present and future diminish, allowing the mind to rest in an open, pure awareness of the here and now [97,161]. This, in turn, leads to a subsequent reduction in the predictive information processes necessary for the construction or simulation of internal models of the self in relation to the world, which is associated with a reduction in DMN activity [97,161,162]. Distorted predictive processes that lead to negative expectations, prediction errors and biased learning have also been implicated in depression, and as such the ability to properly regulate them through augmentation of the DMN via meditation or mindfulness may be crucial for improving depressive symptoms [163,164]. Moreover, a meta-analysis investigating the functional neuroanatomy of meditation by Fox and colleagues (2016) [162] found consistent deactivation within two major hubs of the DMN: the PCC and posterior inferior parietal lobule (IPL). These areas, which have been implicated in mind wandering, episodic memory retrieval, simulation of future situations and conceptual processing, were found to deactivate as a result of FA mindfulness meditation [162].

FA mindfulness meditation is thought to diminish mind wandering (i.e., lack of attentional stability) and spontaneous cognitions related to the past and future by allowing the mind to focus on the present moment without being lost in conceptual elaboration [161,165]. Research suggests that both the DMN and CCN are affected when individuals engage in this type of mindfulness training [119]. Using fMRI, Hasenkamp and Barsalou (2012) [166] observed that individuals engaged in FA mindfulness meditation alternate through four main cognitive cycles, which include mind wandering, awareness of mind wandering, shifting of attention back to the focus of meditation and sustained attention. Mind wandering activates regions associated with the DMN (e.g., dorsal ACC and anterior insula), while shifting of attention and sustained attention activate regions associated with the CCN (e.g., the fronto-parietal executive network and attentional subnetwork) [166]. In this way, continual training can lead to enhanced functional connectivity in both circuits.

Moreover, mindfulness meditation has been found to affect the CCN and DMN via increased activity in the DLPFC [119]. For example, Brewer et al., (2011) found that experienced meditators showed decreased DMN activity (e.g., decreased activation in the medial PFC and PCC) along with increased functional connectivity between regions involved in cognitive control and self-monitoring (e.g., increased coupling between the PCC, dorsal anterior cingulate and DLPFC cortices) while engaged in various types of meditation, as well as when resting at baseline [119]. This altered resting-state activity, however, was only observed among the meditators, suggesting that lasting functional alterations are brought about via long-term meditation [167]. It should be noted that NDA meditation, which seeks to dissolve the boundaries between the objective external environment and subjective internal states, attenuates the anti-correlated activity normally observed between externally oriented networks (e.g., CCN) and internally oriented networks (e.g., DMN) [70]. As such, this would suggest that NDA meditation can dissolve the experience of subject–object duality and potentially lead to more rapid improvements in depression.

## 5. The Next Step: Augmenting CCT with Mindfulness

Given the affective and cognitive deficits associated with depression, an intervention addressing both domains may provide added benefit [16,17,19,20]. CCT and mindfulness have both been shown to be effective interventions in addressing these deficits [45,47,55,56,91]. While CCT enhances the cognitive processes involved in affective regulation, such as inhibition, attention, working memory and planning, mindfulness meditation may provide an additional benefit in the form of psychological training designed to directly counteract difficult emotions (i.e., improve ER) [58,59]. Furthermore, the neural networks implicated in these deficits, the CCN and the DMN, have been found to be augmented by these interventions [119,124,125].

Although both CCT and mindfulness appear to influence the CCN, mindfulness has consistently been found to be associated with changes in the DMN [57,60,70,102,119]. Specifically, CCT seems to lead to improvements in cognitive deficits through upregulation of the CCN, whereas mindfulness leads to improvements in affective deficits via the downregulation of the DMN. Based on these findings, we propose a biologically plausible model of MCCT as a complementary or adjunct treatment for depression that incorporates both CCT and mindfulness (see Figure 1 for details). We predict that following MCCT, individuals will demonstrate increased resting-state functional connectivity and task-based activation within the CCN, and anti-correlated decreased resting-state connectivity and task-based activation within the DMN associated with clinical improvements in depressive symptoms and cognitive function. Further, we predict that improvements in EF and PS will be associated with decreased rumination, and that increased ER and mindful awareness will be associated with decreased negative self-referential processing. These alterations, in turn, will lead to improvements in cognition and depression. Lastly, we hypothesize that the greater anti-correlation observed between the networks will be predictive of depression symptom severity and cognitive improvements.

### 5.1. Feasibility

There is currently no scientific consensus regarding the minimum amount of time required for a regimen of CCT or mindfulness training to promote therapeutic benefits. Across multiple successful studies of adults with depression, CCT training times have ranged from as little as 90 min over 2 weeks [168] to 1000 min over 10 weeks [169], while others have utilized more intensive treatments of up to 1800 min over a 4 week period [51,91]. Meditation programs and MBIs likewise vary, with interventions ranging from as little as 10 min/day over a 10 day period [115] to 47 min/week over a 16 week period [116]. Further, training protocols utilized in CCT and mindfulness research are highly variable. The variability observed in the training durations and protocols required for CCT and mindfulness regimens to demonstrate affective, cognitive and functional improvements seems to suggest the need for a systematic study of the minimally effective doses, training schedules and protocols. In particular, minimum time requirements and ease of use can be understood as important factors affecting treatment compliance and overall feasibility, which consistently pose major issues in both clinical trials and interventions.

Both CCT and mindfulness interventions can be practiced on smartphones, which supports the feasibility of studying and implementing them, given that this format is in line with current trends. The age of the smartphone has now dawned; as of 2020, nearly 85% of Americans own a smartphone [170]. Smartphones contain a diverse and ever-expanding array of applications capable of performing a multitude of functions. These include applications designed to track, monitor and provide interventions for physical and mental health [171,172]. In particular, advancements in smartphone capabilities are transforming the ways in which individuals interact with healthcare providers, implement digital therapeutics and monitor treatments [173,174]. For example, in relation to the social distancing guidelines attributable to the COVID-19 pandemic, telehealth or digital medicine has now become a mainstream and commonly accepted form of intervention [175,176,177]. Based on these trends, it seems probable that the future of healthcare will continue to move in the direction of personalized digital medicine that is readily available to anyone with access to these increasingly ubiquitous mobile technologies [178,179]. As both CCT and smartphone-based mindfulness training programs can be downloaded as apps onto a smartphone and practiced from the comfort of one’s home, they serve as practical and convenient options that support the feasibility of studying and implementing MCCT.

Despite their convenience, it is important to note and consider that dropout rates in clinical trials of smartphone apps for depression have been found to be as high as 50%, making retention a critical issue to consider when planning a study [180]. Although treatment non-compliance is a commonly encountered issue with medication and other interventions, CCT has been made more engaging, fun and user-friendly in order to increase compliance. Unlike medication in which compliance is more difficult to monitor, CCT and smartphone-based MBIs allow compliance and progress to be tracked and monitored remotely.

Based on the high dropout rates observed with remote training applications for depression, it is important to consider whether mindfulness practice without technology, compared to CCT or MCCT, will be more effective for increasing adherence rates and mitigating study dropout. It may be that some individuals will adhere more to mindfulness training when it is guided and structured on their smartphone, while others may prefer a break from technology. However, when it comes to MCCT or CCT, these types of training modalities require the use of technology and may be more attractive and relevant for individuals already more interested, comfortable and engaged with technology. These questions require further investigation.

### 5.2. Transfer of Benefits

The transfer of benefits refers to subsequent functional improvements observed in actual life that carry over and generalize to other skills or cognitive capacities (e.g., faster reaction times in traffic, improved working memory in day-to-day tasks), not only higher scores and improvements in the specific skills trained [181]. The levels of efficacy related to the benefits of CCT have been divided into 4 categories of transfer: (1) training engagement (improved performance on tasks trained); (2) near transfer (improved cognitive performance on non-trained tasks; (3) far transfer (improved performance on cognitively demanding functional tasks); (4) environmental transfer (improvements in everyday functioning) [44]. CCT has been found to have near, far and some environmental transfer of benefits in cognition and functional improvements in daily activities outside of the training sessions [44,92,182]. Despite evidence of far transfer on cognition and function, when a person is no longer engaged in training or involved in an ordinarily demanding cognitive task, they may be confronted with the resurgence of their own negative thoughts and emotions associated with depressive symptomatology. For a depressed individual, self-rumination can be temporarily suppressed when engaged in CCT; however, this suppression ceases when they disengage. As such, CCT alone does not provide a direct remedy to help people deal with negative emotions.

On the other hand, mindfulness can be added to CCT and utilized to help individuals outside of the training context. By becoming familiar with one’s own thought processes and negative emotions, an individual may be able to experience the continued benefits of improved cognitive function (due to CCT) and decreased feelings of depression (due to improved ER ability from mindfulness) throughout their daily activities [60,101,102].

Moreover, altered functional connectivity observed among experienced meditators during non-meditative resting states may be representative of enhanced affective and cognitive abilities being transferred over to daily life after formal training periods [102]. Lastly, decreased DMN activity has been observed to continue after practicing meditation, beyond engagement in the actual training period, suggesting a transfer of psychological benefits and improved ER into daily life [60]. Future research should continue to explore this lasting transfer of benefits outside of the research context.

## 6. Future Directions for MCCT

The next steps in the development and study of mindfulness-enhanced CCT (MCCT) for depression could be through an RCT that assesses cognitive, clinical, functional and neuroimaging variables of interest. These may include measures of EF and PS, depression and CCN and DMN BOLD signal activation and connectivity. Different types of mindfulness and meditation techniques should also be tested in combination with CCT in order to explore differential effects on anti-correlated DMN network connectivity and associated subjective experiences [70].

Within a research context, there are many important questions that need to be addressed to most effectively study MCCT and that are relevant to both basic science and clinical research on mindfulness [59]. Areas requiring further consideration include how to approach the combination of mindfulness and CCT, the modality of training, the training schedule and what the ideal comparator or control conditions would be. As the time required to observe an effect varies from study to study, it would be pertinent for future research to test different durations of MCCT among participants to better understand how the dose level impacts mood and neurocognitive factors among different age groups, genders, depression subtypes and other relevant demographics. Based on the results, a more nuanced understanding of MCCT including its efficacy and mechanisms of action may be deduced. If MCCT is determined to be efficacious for treating depression and its mechanisms of action are more thoroughly studied, it can then be further refined, applied and tested among individuals with different psychopathologies.

Many possibilities and potential applications for MCCT are on the horizon. For example, various forms of CCT, such as ‘closed-loop digital meditation’, utilizing neurofeedback personalized to your brain activity and performance and artificial intelligence deep learning capable of adapting CCT and mindfulness for each individual, are beginning to emerge [160]. The future treatment may be more immersive and may engage the body as well as the mind in order to increase cognitive and affective benefits and overall psychophysical activation. Augmented reality may be used to engage a user in their daily life, allowing for enhanced CCT and mindfulness training in daily activities. This may also increase the transfer of benefits into improved functional activity in daily life, not only improved CCT performance or mindfulness during a formal session. This type of external validity is very important for demonstrating that CCT and mindfulness can actually improve instrumental activities of daily living and mood.

## 7. Conclusions

Based on current trends in the proliferation of mindfulness training apps, the rise of CCT programs with enhanced user interfaces, the burgeoning interest in mindfulness meditation and the exponential growth in the use of smartphones globally, augmenting CCT with mindfulness (MCCT) is a logical and promising frontier to explore in depression research. The enhancement of cognitive function through CCT combined with the enhancement of affective regulation through mindfulness has the potential to boost treatment response in MDD. In conjunction with pharmacological or psychotherapeutic treatments for MDD, targeting the CCN and DMN via MCCT may act to improve cognitive and affective functioning more rapidly compared to standard CCT. Moreover, mindfulness training provides an additional therapeutic tool that depressed individuals can incorporate into their day-to-day activities, not only when engaged in the use of a cognitive training app. Based on the substantial parallel lines of research related to the effects of CCT and mindfulness on these two interrelated neural networks, MCCT appears to be a biologically plausible adjunctive intervention that builds upon previous theory. For these reasons, MCCT is an important and promising new direction for CCT research requiring further exploration.

## Figures and Tables

**Figure 1 brainsci-12-00663-f001:**
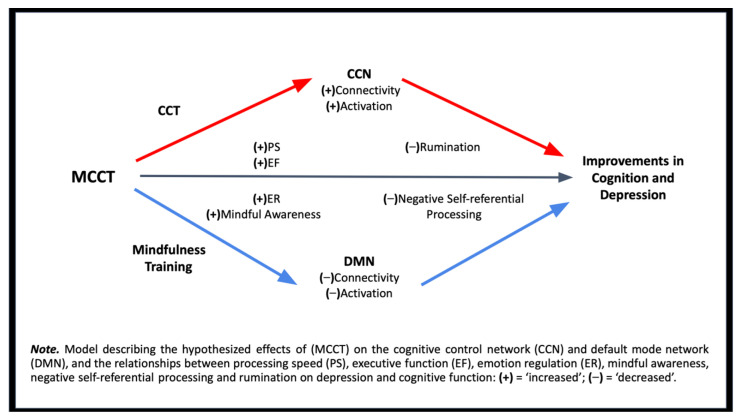
Mindfulness-enhanced computerized cognitive training (MCCT).

## Data Availability

No new data were created or analyzed in this study. Data sharing is not applicable to this article.

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
