# Peer review of "Mindfulness-Enhanced Computerized Cognitive Training for Depression: An Integrative Review and Proposed Model Targeting the Cognitive Control and Default-Mode Networks"

_brainsci, 2022, doi:10.3390/brainsci12050663_

Round 1

Reviewer 1 Report

The authors reviewed the recent literature on the efficacy of Computerized Cognitive Training (CCT) and mindfulness training for treating depression, leading to the proposal of neural mechanisms associated with the effect of a novel intervention that combines these two forms of training. The manuscript presented a comprehensive overview of the topic and relevant well-rounded studies. The proposed model of the neural networks involved in the efficacy of Mindfulness Enhanced CCT (MCCT) for treating depression is also novel and interesting.

The following minor issues may need to be addressed before full acceptance for publication:

  1. In line 66, it says “Further, the effects of psychotherapy without the combination of medication may be overestimated or ineffectual (Cuijpers et al., 2010).” However, does “overestimated effects” mean that the treatment effects are not significant? In addition, have there been any studies that compare the effects of CCT and those of traditional psychotherapy or medications for depression? I would suggest that the authors elaborate on the advantages of CCT (e.g., time saving, easier access) over traditional forms of psychotherapy in the paragraph from line 78 to line 89.   

  1. In line 85-86, it indicates “Although CCT targets aspects of cognition, it does not directly target affective dysregulation or negative self-referential processes which are hallmarks of depression.” However, in section 2. Depression and Computerized Cognitive Training, it seems to suggest that one key component in the CCT effect for depression is the improved emotion regulation through enhanced PS and EF. It would be helpful if the authors could elaborate a bit more concerning the relative disadvantages of CCT in improving affective dysregulation in depression as compared to mindfulness training.

  1. I anticipate that when reading the current paper, some audience may think of the Mindfulness-Based Cognitive Therapy (MBCT) developed by Zindel Segal et al. as an increasingly popular treatment approach for treating depression, which is also a combination of cognitive therapy and mindfulness training. There has also been established computerized MBCT program (e.g., Dowd, H. D. et al. [2015], Comparison of an online mindfulness-based cognitive therapy intervention with online pain management psychoeducation, The Clinical Journal of Pain, 31(6), 517-527). To improve the novelty of this manuscript, I would suggest that the authors add discussions on the distinctions between the MBCT and the proposed MCCT program for treating depression.

  1. There has been literature on the anti-correlated activity in CCN and DMN and decreased functional connectivity between CCN and DMN in MDD patients (e.g., Yao, Z. et al. [2019]. Altered dynamic functional connectivity in weakly-connected state in major depressive disorder, Clinical Neurophysiology, 130(11), 2096-2104.). Do the authors have any predictions of the relations between the post-MCCT alternations in the CCN and the DMN activity?

  1. In line 502-503, it says that “CCT is limited in the extent to which it can improve depression outside of the training context”. Please cite relevant literature in support of this statement.

Figure:

  1. Please cite the sources of the two brain figures in Figure 1. In addition, I found the details of the brain figures very difficult to read. Overall, the quality of the figures needs to be improved. For example, the resolution of the figures seems quite low. The authors can refer to some online resources (e.g., the Purdue Online Writing Lab) for creating production-quality figures.

  1. In Figure 1, lease specify in which direction that CCT/mindfulness may modulate the activity in CCN/DMN, and the roles of changes in the neurocognitive/ER-relevant processes (e.g., PS, EF, negative self-referential processing) in these pathways.

  1. It appears that Figure 1 and 2 share a lot of overlapping contents. To reduce redundancy, I would suggest that the authors consider combining these two figures into one.

Reviewer 2 Report

Overall the paper is well-written with strong conceptualization from literature review to definitions and future directions. The description of MDD is succinct and ease to understand to even readers less familiar with the nuances. The reason for the study is clear and evidenced with the statistics in the opening. The case is well made that this is a next step for research with important variables noted. The figures are very helpful and affective. See a few comments below re: three recommendations for minor additions to strengthen the manuscript. 

Pg. 2 Line 84: "Although 84 controversy surrounding its use exists (Harvey et al., 2018)" - I would recommend a sentence or 2 noting the controversy. Though the citation is provided for ethical reasons the reader should be provided with this information.

Pg. 4 Lines 184-188 - As the reader it took some effort to decipher if the spectrum was in each of the 3 domains or FA, OM, and NDA. The concepts and continuum do not obviously map and reader with less familiarity with mindfulness could benefit from more clarity on these points. Also while the first to concepts are easily understood based on the mindfulness definition non-duality is not defined or explained. Adding a definition here or in the explanation of mindfulness would strength reader understanding. 

Pg. 10 Section 5.1 Feasibility - Important information and research about the use of apps and accessibility of the CCT and mindfulness. I feel it would be important in the introduction or early in the paper to give a brief description of what CCT looks like and what mindfulness training entails. Readers may not be familiar and experts may be familiar with one but not both. These examples or descriptions could be provided in the text or as another figure for reader reference. I saw CCT described with a more concrete example in the future directions section but having it at the beginning with more description would help the reader track with the argument. 

Finally this may be a journal directive and not an error, but I notice "and" between two authors names in-text citations versus the typical APA style of using "&". 

Reviewer 3 Report

The integrated review of CCT in depression is an interesting idea. The scope of this manuscript is too broad and doesn't allow to describe the nuances involved and many areas unexplored. Consider revising the manuscript to two goals rather than the present two goals.

Round 2

Reviewer 3 Report

The authors attention to comments and revisions are appreciated. One more point of clarification would be appreciated. Line #70 Psychiatric medication such as selective serotonin reuptake inhibitors (SSRIs) ; Hyttel, 1994) is the first line of treatment for MDD. Hyttel 1994 doesn't support this statement. The paper indicates SSRIs are more commonly used compared to TCAs. The current recommendation is that medication + psychotherapy. Some differences in first line of medication classes for mild/mod depression vs severe depression (i.e., SSRI vs. SNRI). 
